# Metabolic Fatty Liver Disease in Children: A Growing Public Health Problem

**DOI:** 10.3390/biomedicines9121915

**Published:** 2021-12-14

**Authors:** Sébastien Le Garf, Véronique Nègre, Rodolphe Anty, Philippe Gual

**Affiliations:** 1Centre Spécialisé de l’Obésité PACA Est, Pôle Digestif-Anesthésie-Réanimation-Endocrinologie (DARE), CHU, Membre de l’Université Côte d’Azur, 06204 Nice, France; sebastien.LE-GARF@univ-cotedazur.fr (S.L.G.); negre.v@chu-nice.fr (V.N.); 2Université Côte d’Azur, CHU, INSERM, U1065, C3M, 06204 Nice, France; anty.r@chu-nice.fr; 3Université Côte d’Azur, INSERM, U1065, C3M, 06204 Nice, France

**Keywords:** NAFLD, MAFLD, pediatric, obesity

## Abstract

Metabolic-associated fatty liver disease (MAFLD), previously called nonalcoholic fatty liver diseases (NAFLD), is one of the most important causes of chronic liver disease worldwide and will likely become the leading cause of end-stage liver disease in the decades ahead. MAFLD covers a continuum of liver diseases from fatty liver to nonalcoholic steatohepatitis (NASH), liver fibrosis/cirrhosis and hepatocellular cancer. Importantly, the growing incidence of overweight and obesity in childhood, 4% in 1975 to 18% in 2016, with persisting obesity complications into adulthood, is likely to be harmful by increasing the incidence of severe MAFLD at an earlier age. Currently, MAFLD is the leading form of chronic liver disease in children and adolescents, with a global prevalence of 3 to 10%, pointing out that early diagnosis is therefore crucial. In this review, we highlight the current knowledge concerning the epidemiology, risk factors and potential pathogenic mechanisms, as well as diagnostic and therapeutic approaches, of pediatric MAFLD.

## 1. Introduction

Obesity is a worrying disease, both in terms of its growth around the World and the associated complications such as metabolic disorders that affect numerous organs, such as the liver [1]. The liver diseases associated with metabolic syndrome, initially called nonalcoholic fatty liver diseases (NAFLD), have recently been renamed, in adults, metabolic-associated fatty liver diseases (MAFLDs) [2]. Although the term NAFLD was well established in the international literature, this change in terminology in adults was recently motivated by several reasons. The main reason was that the name of this liver disease should better reflect the cause(s) of the disease, and the second was to remove the term “nonalcoholic”. The term “nonalcoholic” was explained by the genesis of the identification of the disease over forty years ago. However, the term was sometimes misunderstood by adult patients. An expert group has thus proposed “metabolic (dysfunction)-associated fatty liver disease” (MAFLD). Curiously, instead of closing the previously existing debate on the imperfection of the term “NAFLD”, many experts have criticized the new definition of MAFLD. Among the main criticisms is the absence of exclusion of viral hepatitis B or C or of a threshold for alcohol consumption. Although this “type” of adult patients is regularly seen in “real life”, this adds heterogeneity to the definition of patients. This new definition could increase the complexity instead of simplifying the concept. In adults, recent studies have now compared the different characteristics of NAFLD+/MAFLD- patients, NAFLD-/MAFLD+ patients and NAFLD+/MAFLD+ patients [3]. These difficulties may be due to the imperfect classification according to qualitative thresholds of metabolic abnormalities that appear progressively over time and in a variable manner among patients. The criteria used for defining metabolic abnormalities in MAFLD, although easy to use in clinical routine, may be somewhat simplistic.

In this review, and far from the controversies of experts, MAFLD has been used to designate chronic liver disease related to overweight/obesity, chronic low-grade inflammation and insulin resistance, as some experts have recently suggested [4,5]. Moreover, in children, possible alcohol-related liver disease is unlikely.

After clarifying the use of this new terminology, it is also important to emphasize the importance of eliminating other etiologies of liver disease not related to metabolic damage. In young children under 10 years of age, fatty liver disease is rather uncommon. The causes of this liver steatosis related to specific metabolic abnormalities should be investigated as a priority, including the presence of congenital metabolic diseases (e.g., deficiency of α1-antitrypsin, cystic fibrosis, galactosemia, tyrosinemia type 1, defective peroxisomal β-oxidation), since MAFLD in very young children seems to be relatively rare [4]. 

MAFLDs are the most common chronic liver diseases in the world, with a global adult prevalence of 25% [6]. The main comorbidities associated with obesity are risk factors for the progression of MAFLD [7] and vice versa. Indeed, MAFLD is a risk factor for many metabolic diseases such as cardiovascular disease (the main cause of death worldwide [8,9]) and type 2 diabetes mellitus. Age, duration and severity of obesity also influence the prevalence of MAFLDs and their gravity. Indeed, the prevalence of imaging-defined MAFLD in severely obese adults with coexisting metabolic syndrome features is approximately 90–95% and more than one third of these patients have nonalcoholic steatohepatitis (NASH) on histology [10]. MAFLD encompasses a continuum of liver diseases, ranging from liver steatosis to metabolic steatohepatitis, hepatofibrosis and hepatocellular cancer. The known causes and risk factors of MAFLD are closely linked to modernization of lifestyle (dietary changes and reduced physical activity associated with socioeconomic factors). Obesity, type 2 diabetes and MAFLD share a partial common pathophysiology including, in particular, metabolic inflammation and insulin resistance. These anomalies are themselves associated with systemic complications, such as atherosclerosis and extrahepatic cancers. 

Childhood obesity has emerged as an important public health concern worldwide. In fact, the prevalence of overweight/obesity among youth aged 5–18 in the world has perilously risen from just 4% in 1975 to 18% in 2016. Even more worrying is the fact that 39 million children under the age of 5 were overweight or obese in 2020 (WHO 2021), with a preponderant risk of complications. For example, children with obesity have a fourfold increase in the risk of developing type 2 diabetes than children with a “normal” body mass index (BMI) [11]. The pathophysiology of excess weight gain is complex with the interplays between epigenetic and genetic factors, physical and social environment, and biological factors [12]. The growing prevalence of childhood obesity is closely related to the development of comorbidities previously considered diseases observable only in adulthood, such as MAFLD (Figure 1). This review will provide overall picture of the current prevalence, diagnosis and understanding of MAFLD including noninvasive approaches, molecular mechanisms, and potential therapeutical approaches in pediatric obesity.

## 2. General Information on Pediatric MAFLD

### 2.1. Risk Factors and Epidemiology

The epidemiologic data in this population are alarming and track obesity level like those in their elders. From general population studies, MAFLD prevalence in children is between 7.6–9.6%. From studies based on child obesity clinics, this prevalence is 34.2%. Globally, MAFLD and NASH prevalence augmented from 19.34 million in 1990 to 29.49 million in 2017 in young people, with an incidence that increased to 1.35. The greatest increase was in the Middle East and North Africa. MAFLD prevalence in children also varies by ethnicity. Children of Hispanic (11.8%) and Asian (10.2%) ethnicities have a higher prevalence of MAFLD compared with Caucasian children (8.6%). It is also important to consider the gender component. The prevalence is generally higher in boys (9% compared to 6.3% in girls in the general youth population) and progressively rises with greater BMI (35.3% vs. 21.8% in obese boys vs. girls) [13]. In addition, the prevalence of NASH is also of concern, involving 23% of children with MAFLD, of which 9% showed advanced fibrosis or cirrhosis, indicating that the disease can be progressive even in childhood [14]. A plausible link between the pathogenesis of MAFLD in the adult and child populations is lifestyle (e.g., high prevalence of high-fat and nutritionally poor diets, reduced physical activity and increased sedentary behavior) [15].

### 2.2. Histological Differences between Pediatric and Adult MAFLD 

From a liver biopsy, which is still the gold standard, the hepatic histopathological analysis evaluating hepatic steatosis, NASH and fibrosis has been well described for children. The typical pattern in adult (named type 1 NASH) displays the presence of steatosis with hepatocellular ballooning and/or perisinusoidal fibrosis (zone 3 lobular contribution) and relative sparing of the portal tracts. Interestingly, differences in the pediatric population have been reported (Figure 2). For example, the pericentral steatosis (zone 3) is more related to steatohepatitis, whereas the periportal steatosis (zone 1) is more related to fibrosis. In addition, pediatric NASH (called type 2 NASH) has been defined by the presence of steatosis with portal inflammation and/or fibrosis without perisinusoidal fibrosis and hepatocellular ballooning. However, the overlapping features of both types (1 and 2) have often been reported in pediatric patients [16]. The causes of these differences between adults and children are still poorly understood. However, it has been suggested that genetics could partially explain this difference of NASH patterns among children [17]. This singular feature in children underlines the difficulty of understanding the pediatric MAFLD pathogenesis. Most children with MAFLD have been diagnosed between the ages of 10 and 13. At the time of diagnostics, 10–25% of them have advanced steatosis (NAFL) and 20–50% have NASH [18]. Due to the disadvantages and risks associated with liver biopsy and as pediatric MAFLD represents a potential future major public health concern, it is essential to develop new noninvasive methods to diagnose, grade and stage MAFLD.

### 2.3. Pathogenesis of Pediatric MAFLD

Liver steatosis, the hallmark feature of MAFLD, is mainly the consequence of the imbalances in hepatic lipid homeostasis when free fatty acids (FFA) from adipose tissue (AT) lipolysis and de novo lipogenesis exceed the liver’s ability to secrete excess lipids via very low density lipoproteins and/or to oxidize them. This metabolic dysregulation is caused by a complex interaction between environmental, genetic and epigenetic factors. In the adult population, NASH, the progressive form of MAFLD, is the consequence of abnormal activation of liver conventional immune, endothelial and parenchymal cells by inflammatory mediators from the gut, adipose tissue (AT) and liver. The secretion of inflammatory mediators by hepatocytes, resident macrophages (Kupffer cells) and liver sinusoidal endothelial cells, for example, strongly contributes to the substantial liver accumulation of neutrophils and bone-marrow-derived macrophages [19,20]. However, this pathogenic event is less clear in children. Since the diagnosis of MAFLD in children usually occurs during or after puberty, particular caution in interpreting the presence of insulin resistance should be considered. While this insulin resistance is a marker and an actor of MAFLD in adults, the insulin resistance in teenagers could be more related to puberty than to MAFLD. Indeed, puberty is often accompanied by a physiological disruption of insulin pathways leading to fat accumulation to acquire the adult phenotype [21]. While insulin resistance (IR) may predict MAFLD, interactions among IR, fat deposition differences, cytokine/adipokine profiles and sex hormones may all contribute to the gender differences observed in MAFLD in the youth population. Changes in mechanisms regulating the homeostasis of the liver and extrahepatic organs and the “stability” of gene and epigenetic mechanisms are important pathogenic factors in MAFLD development. 

### 2.4. Intra and Extrahepatic Mechanisms

In the adult population, a fat overload into the liver (such as free fatty acids) exacerbates IR by impairing, for example, the phosphorylation of the major insulin receptor substrates by insulin receptors [22]. Hepatic steatosis is also associated with a decreased level of insulin sensitizer adipokines. Therefore, a major aspect of the imbalance of fat metabolism is the dysregulation of insulin signaling pathways (such as adipokines, FAT/CD36, HSL enzyme, PPARα) and exacerbation of nonesterified fatty acids which leads to cellular stresses and triglyceridemia imbalance. AT has emerged as a decisive and critical player in the development of systemic low-grade inflammation, IR and MAFLD (Figure 3). Augmented lipolysis and secretion of proinflammatory (e.g., IL-6, TNFα, IL-1β) and profibrogenic adipokines (Leptin, resistin, osteopontin, etc.) are well documented. Adipose tissue secretes adipokines which are involved in the appearance and the progression of this disease. The location of adipose tissue is also a determining factor in this pathophysiological mechanism. Indeed, visceral adipose tissue is intimately associated with the increase in leptinemia, inflammatory cytokines and a reduction in adiponectinemia. Adiponectin plays a key role in increasing FFA oxidation and decreasing FFA influx to the liver, gluconeogenesis, and de novo lipogenesis, thus supporting a hepatic protective role. In addition, it also plays a hepatic anti-inflammatory and anti-fibrotic role, by reducing proinflammatory cytokines and the activation and proliferation of hepatic stellate cells. Two main inflammatory pathways, JNK-AP-1 (Jun N-terminal kinase-activator protein-1) and IKK-NF-κB (inhibitor of NF-κB kinase-nuclear factor kappa-light-chain-enhancer of activated B cells), are critically involved in the development of the chronic inflammation occurring during MAFLD [23]. These physiological mechanisms lead to increased cellular stresses (oxidative and endoplasmic reticulum stresses) driving hepatocellular death and NASH onset and progression [24]. These changes in adipose tissue secretagogues promote the onset of insulin resistance linked to the increase in FFA and hepatic lipogenesis. Moreover, alterations of free cholesterol metabolism seem involved in the development and progression of MAFLD and are consequently associated with a higher cardiovascular risk [25]. There is currently less understanding of the molecular mechanisms that may be involved in the genesis of pediatric MAFLD. Indeed, this lack of knowledge is probably related to the different zonation of liver damage (i.e., lobular or portal inflammation) in the young obese population. 

The lasting modification of the intestinal microbiota (intestinal dysbiosis), as a central element in the development of many systemic diseases such as obesity and MAFLD, is well documented in experimental adult murine models and adult patients (Figure 4). However, it is poorly documented in the context of pediatric MAFLD [26]. Since the microbiome affects the utilization of nutrients, immune functions and host gene expression, it likely plays a role as a driver of youth-onset MAFLD. Indeed, the intestinal microbiota contributes to systemic inflammation and thus promotes the progression of hepatic lesions. It should be noted that more than 50% of the blood supply to the liver is provided by the splanchnic vascular network. The liver is therefore one of the organs most exposed to intestinal toxins due to its role as the first wall of protection. In summary, intestinal dysbiosis is associated with altered intestinal metabolites, modulation of the local immune systems, and increased intestinal permeability and translocation of bacterial products (PAMPs including LPS). In line with this, an association between specific microbial patterns and pediatric MAFLD has recently been reported. In a study involving 124 obese children, the children with MAFLD had lower alpha diversity (i.e., abundance of Firmicutes compared with Bacteroidetes) than obese children without MAFLD, and the alpha diversity correlated with MAFLD severity. Furthermore, it has been suggested that specific dietary patterns, such as high-fructose or high-fat diets, may lead to MAFLD in part by mediating dysbiosis and intestinal permeability. This increased translocation of bacterial products has been associated with liver inflammation and MAFLD [27]. 

On the other hand, a change in the microbiome of the offspring in case of maternal obesity or maternal obesogenic diet has been reported in preclinical models and the adult population. The gut microbiota in infants of obese mothers increased inflammation and susceptibility to MAFLD development [28]. The initial “pioneering” bacteria present in the gut appear to teach the immune system specifically innate immunity. Kalliomäkiet al. compared groups of children over time and demonstrated that those who became overweight by the age of 7 years old had already a higher level of *Staphylococcus aureus* and a lower level of *Bifidobacteria* compared to those with a weight in the normal range [29]. Indeed, anatomically, the liver is a true receptacle for bacteria and derived products, filtering them from incoming blood through the hepatic artery and the portal vein. In a recent review [30], studies performed specifically in young people with MAFLD showed an increase in *Prevotella copri*, *Proteabacteria* and *Escherichia* species. Bacteria-derived products such as bacterial RNA and DNA can also induce host inflammatory responses by activating the innate immune defense system such as the inflammasome. Furthermore, in a recent study, 125 obese children with small intestinal bacterial overgrowth (SIBO) had an augmented risk for MAFLD development. The relationship between diet and gut dysbiosis may impact the gut–liver axis [31].

### 2.5. Involvement of Genetic and Epigenetic Factors

The rise of pediatric MAFLD parallels the increased number of obese children; however, not all children with obesity develop MAFLD and a gene–environment interaction is probably at the origin of the susceptibility to the disease [32]. The heritability of MAFLD is estimated to be 35–61% and, in particular, the predictive biomarkers of this disease (e.g., circulating liver enzymes, such as ALT and AST) have a heritability close to 60%. Family studies have also demonstrated that family members of individuals with cirrhosis related to MAFLD have a 12.5-fold increased risk of progressive MAFLD compared to the general population, independently of several confounders [33,34]. Genome-wide association studies (GWAS), candidate gene studies and epigenetic studies provided insight into the role of genetic factors in MAFLD susceptibility and progression. Single nucleotide polymorphisms (SNP) have been related to liver steatosis and the progression of hepatic fibrosis in children. *Patatin-like phospholipase domain-containing protein 3* (*PNPLA3*) polymorphism has been related to the onset of liver steatosis mainly via the impairment of very low density lipoprotein (VLDL) secretion and increased lipogenesis. In children, the *PNPLA3* polymorphism may be a stronger predictor of fatty liver than insulin resistance. Among the other genetic variants associated with pediatric MAFLD risk (Table 1), variants of *Transmembrane 6 superfamily member 2* (*TM6SF2*) gene, which reduces lipid mobilization via the VLDL, have been also reported [16,33]. The current literature emphasizes the need for vigilance in distinguishing susceptibility genes (e.g., *PNPLA3*, *TM6SF2*, *glucokinase regulator* (*GCKR*) and *interferon-γ4* (*IFNL4*)) from those of disease progressions (Table 1). However, variants in *GCKR* and *IFNL4* could have a lower impact on the risk of MAFLD than *PNPLA3* and *TM6SF2* variants. Otherwise, their effect on the risk of pediatric MAFLD is currently not well established [35]. The main other genetic mutations associated with pediatric MAFLD progression are carried on genes coding for nuclear receptors (i.e., *PPARα*, *PPARγ* and *LPIN1*), mitochondrial proteins (i.e., *UCP2*) and proteins involved in insulin signaling pathway regulation (i.e., ectonucleotide pyrophosphatase phosphodiesterase 1 (*ENPP1*) and *IRS1*) [16,36].

MAFLDs are also dependent on the interactions between the environment and a susceptible genetic background of the host. Indeed, epigenetic parameters, which result in gene expression regulation without any DNA sequence changes, could be caused by alterations in DNA methylation and/or deregulation of liver microRNAs (such as miR-122, which is associated with increased lipogenesis), for example [37]. These could contribute to MAFLD development. However, these responses are reversible and could be modified by changes of lifestyle such as physical activity and diet. The impacts of some of these therapeutic strategies on pediatric MAFLD will be discussed further below. As an example, in children with MAFLD, dietary supplementation or enrichment with polyphenols could have beneficial effects by reducing the activity of the liver enzymes and level of inflammatory cytokines [38]. In an experimental study, this supplementation restored the NAD-dependent deacetylase sirtuin-1 (SIRT1) activity. It is well documented that SIRT1, by at least enhancing PPARα activity and fatty acid oxidation, improves the metabolism and insulin signaling [33].

Over the last ten years, several studies have shown that the onset of MAFLD may be influenced during the preconceptional period and the growth of the fetus in utero. Perinatal factors may also impact the MAFLD development during childhood, however their relative contributions to MAFLD severity in adulthood are not yet fully demonstrated. In a recent systematic review, a prepregnancy obesity was clearly related to an increased risk of pediatric MAFLD [39]. This association has been confirmed in a recent Swedish study [40]. In the previous systematic review, pregestational or gestational diabetes and abnormal birth anthropometrics were not definitively associated with pediatric MAFLD. On the contrary, breastfeeding was negatively related to the risk of pediatric MAFLD [32]. In an Australian longitudinal birth cohort including around 2900 offspring, a maternal BMI greater than or equal to 30 before pregnancy was related to an augmented MAFLD risk in offspring [41]. This is even more alarming as the prevalence of obesity and MAFLD in pregnant women continues to rise [42]. In addition, the paternal influence might also increase the MAFLD risk, though few studies are available. Paternal obesity can promote programmed phenotypes in offspring via epigenetics. The human evidence for infant and prenatal risk factors related to MAFLD are clearly detailed in the Mann’s review [43]. Moreover, independently of the prior existence of obesity, diabetes or hypertension, MAFLD in a pregnant woman can cause unexpected deleterious effects by significantly increasing the risk of serious pregnancy-related complications such as hypertension (which increases threefold), postpartum hemorrhages, pre-eclampsia and premature births [42]. Recent data illustrate first-degree transmission of NASH with fibrosis. The respective roles of genetic and epigenetic factors and environmental factors including transmission of gut microbiota from mother to child and dietary and exercise habits remain to be clarified [34,44]. In any case as a preventive measure, acting on epigenetic factors may also be beneficial to health. For example, it has been reported that maternal exercise during pregnancy protects the offspring from HFD-induced weight gain and hepatic steatosis in mice [45]. During this period, physical activity has been shown to improve metabolic health in offspring and confers protection against the development of MAFLD. However, the underlying mechanisms are still poorly understood. In the murine MAFLD model, this behavior is associated with a significant activation of the hepatic AMPK–ACC–CPT1a axis. PPARα and PGC1α signaling pathways increase hepatic β-oxidation and reduce liver lipogenesis. This study concludes that this epigenetic factor (i.e., paternal obesity has been associated with an increase in sperm’s oxidative stress and greater susceptibility of a DNA methylation and histone modifications, which could contribute to a defect in embryonic development) modulates offspring liver function through early hepatic programming of offspring hepatocytes [46].

## 3. Medical Diagnosis

The differential diagnosis of pediatric MAFLD is first based on the exclusion of other causes of chronic liver disease such as viral infection (hepatitis B and C), autoimmune hepatitis, Wilson’s disease, genetic hemochromatosis or biliary disease. Moreover, the research of other causes of liver steatosis is important, particularly below the age of 10, to diagnose inborn errors of metabolism. The latter constitute a large category of genetic diseases that involve congenital disorders of enzyme activities (e.g., alpha1-antitrypsin deficiency, type 1 tyrosinemia, galactosemia, defect in mitochondrial and peroxisomal fatty acid oxidation, cystic fibrosis). 

### 3.1. Liver Biopsy

The most robust method to assess MAFLD (steatohepatitis) is the liver biopsy; however, it is used clinically as a last resort and only for the most advanced and severe forms, as pointed out in the review by Castillo-Leon et al. [32]. However, this approach is still a reference for identifying progressive forms of MAFLD, especially in children, where MAFLD progresses differently than in adults, with potentially serious repercussions in adulthood. In general population studies, MAFLD prevalence has been shown to differ between diagnostic methods. In a clinical study on obese children and teenagers, estimates of the prevalence of MAFLD were similar between ultrasound scan (USS)and MRI methods but were highly underestimated compared to liver biopsy. This illustrates the difficulty of the noninvasive methods to accurately diagnose MAFLD in very obese individuals [13]. This lack of biopsy use in children/adolescents with suspected advanced MAFLD is a real challenge for the prevention of severe forms of MAFLD. Indeed, much less is known about the prevalence of steatohepatitis or fibrosis in the child population. Therefore, there is an urgent need to develop new technologies to noninvasively assess these disease endpoints and avoid MAFLD progression. This is essential because the majority of cases of cirrhosis in young adults could be the outcome of undiagnosed MAFLD in youth. The Executive Council of the North American Society for Pediatric Gastroenterology, Hepatology, and Nutrition (NASPGHAN) recommended a liver biopsy in children at risk of NASH and/or advanced fibrosis, considering risk factors such as the presence of type 2 diabetes, splenomegaly, ALT > 80 U/L, AST/ALT > 1 and disease-related panhypopituitarism. When a liver biopsy is performed in these cases, approximately one out of four children display NASH or borderline NASH, and 15% display stage 3 hepatic fibrosis [15]. While the liver biopsy is necessary for the definitive diagnosis of NAFL/NASH, this clearly cannot represent a screening procedure. However, liver biopsy in children with suspicion of MAFLD should be performed when the diagnosis is uncertain, when there is a possibility of several diagnoses, or before starting therapy with hepatotoxic medications [47].The first line of screening should be the use of noninvasive tests as soon as a child has high-risk factors (i.e., maternal obesity, (pre)gestational diabetes, increased prepregnancy obesity, family history of first-degree NASH, abnormal birth anthropometrics). 

### 3.2. Noninvasive Methods

#### 3.2.1. Simple Serum Biomarkers

The most common serum biomarker used to assess MAFLD prevalence is the marker of liver injury alanine aminotransferase (ALT). Based on the NASPGHAN guidelines, further evaluation is necessary if ALT levels are ≥ 80 U/L at the first visit, or if ALT levels are persistently elevated (>3 months) to twice the upper limit of normal ALT levels (in girls ALT ≥ 44 U/L, in boys ALT ≥ 52 U/L). Interestingly, it has been reported that the BMI z-score (also called BMI standard deviation score, a measure of relative weight adjusted for child age, height and sex) positively correlated with ALT serum concentrations. An increase in the BMI z-score of around +1 was associated with an increase in ALT serum level of around 2.2 U/L in boys and approximately 1.1 U/L in girls [48]. As indicated in Clemente’s review [24], although MAFLD are the most frequent cause of elevated transaminase in obese children and teenagers, elevated ALT is not a sensitive marker to predict liver disease and its severity. According to the “Screening ALT for elevation in today’s youth” (SAFETY) study, normal transaminase values for teenagers and children are currently set too high relative to the threshold for detection of fatty liver. The 95th percentile levels for ALT in healthy weight, metabolically normal children, free of liver disease, should be 25.8 U/L for boys and 22.1 U/L for girls. With this cutoff, the sensitivity of the diagnosis increased from 32% to 80% in boys and from 36% to 92% in girls. In addition, joint assessment of AST and ALT values is essential because an elevated AST/ALT ratio may reflect progressive and more serious liver disease, such as fibrotic NASH. 

In addition, metabolic assessment (blood glucose, insulin levels, insulin resistance (via the homeostatic model assessment of insulin resistance) and lipid profile) should also be conducted in all children with suspected or diagnosed NAFLD. However, these systemic indicators for MAFLD screening remain controversial and specific thresholds should be established according to age, sex and/or ethnicity. Indeed, these may underestimate the prevalence of MAFLD in young obese people and overestimate it in the general population. The place of these noninvasive markers in the diagnosis of MAFLD in children remains to be clarified, supported and finally validated in usual clinical practice. In addition, combining these different parameters in scoring models could be more useful.

#### 3.2.2. Imaging

##### Imaging for the Diagnosis and Quantification of Steatosis (USS, CAP, MRI PDFF)

The use of ultrasound scanning (USS) for screening has been recommended by the European Society for Pediatric Gastroenterology, Hepatology and Nutrition (ESPGHAN) in 2012. USS is the most widely used imaging technique to estimate the prevalence of MAFLD in the general population. Conventional ultrasound is mainly used for the qualitative estimation of steatosis. USS predicted moderate to severe steatosis with a good estimated sensitivity (85%) and specificity (94%) in a cohort of 208 children with biopsy-proven MAFLD [49]. USS has been recommended for obese children screening with signs of insulin resistance and/or elevated transaminases [50]. However, the major limitations of USS are indirect measurement of the fat content, and subjective and mainly nonquantitative examination. New techniques for quantifying steatosis coupled with ultrasound machines are being developed and may be of interest in the future [51].

Controlled attenuation parameter (CAP) has been incorporated in the transient elastography (TE) system of FibroScan^®^. The main advantage of the CAP technique is the quantification of hepatic steatosis diagnosed by ultrasound, although several limitations remain. For example, some experts said that quantification of steatosis had little or no value in clinical routine, since the severity of MAFLD was essentially related to the severity of fibrosis. Moreover, there is no consensus on thresholds for defining the severity of steatosis in adults or children. However, CAP reached high specificity and sensitivity (80% and 98.7%, respectively) using a size S probe to assess the absence and presence of steatosis in a total of 86 children with MAFLD [52]. On an individual basis, improved CAP values could help in monitoring therapeutic management with drug or nondrug therapeutic measures. Lowering CAP could also be of interest in terms of patient motivation. Improvements in hepatic steatosis have been described even in the absence of weight loss in adult MAFLD patients undergoing a sustained physical activity program. Recently Echosens^®^, the company that markets CAP, has proposed a new application called the “Continuous CAP” that would give more reliable values.

Magnetic resonance spectroscopy–proton density fat fraction (MRS-PDFF) and magnetic resonance imaging-PDFF (MRI-PDFF) have proven to be very precise and robust methods (and better than CAP in adults), but their use in routine clinical practice remains limited due to their low availability in public institutions and the relative complexity of acquisition [53,54].

##### Imaging for the Diagnosis of Cirrhosis and Portal Hypertension

USS, computed tomography (CT) scans or MRI may be useful to diagnose advance liver disease characterized by specific signs of chronic liver disease (e.g., dysmorphic liver, with the presence of bumpy contours, ascites, etc.) and/or portal hypertension (e.g., porto-systemic venous shunts, enlarged spleen, etc.).

#### 3.2.3. Noninvasive Tests Used to Diagnose and Stage Liver Fibrosis 

##### Serum Markers

To date, multiple scores have been proposed to predict and estimate the degree of fibrosis. Some of them are open access and can be easily determined with specific calculators (Table 2) [55]. The AST to platelet ratio (APRI) and fibrosis-4 (FIB-4) indexes were designed on adults, and their performance on pediatric subjects has been inconsistent, with areas under the receiver operating characteristic curve (AUC) that range from 0.67 to 0.70 and from 0.54 to 0.81 for APRI and FIB-4, respectively. More specific scores to predict fibrosis in children have been proposed including the Pediatric NAFLD Fibrosis Index (PNFI) and Pediatric NAFLD Fibrosis Score (PNFS). These scores obtained higher values of AUC in training cohorts of 203 children with MAFLD (0.74 and 0.85 for PNFS and PNFI, respectively) [16,56]. The PNFS better predicted advanced fibrosis in children and fibrosis related to advanced liver disease (i.e., NASH type 2 as described previously) [53]. People with MAFLD in childhood are at greatest risk for more severe complications later in life. Therefore, an effective and preventive screening strategy to identify children with MAFLD is needed, especially because children who develop MAFLD may be asymptomatic, with few or no signs of the metabolic syndrome. 

Other technologies assessing fibrosis, such as Fibroscan^®^ VCTE (vibration-controlled transient elastography) and Fibrotest^®^, are available but mainly validated in youth with viral hepatitis (B or C), or other pediatric liver diseases. While a validation in pediatric MAFLD has to be performed, a study on a cohort of 116 children with chronic liver disease showed that the assessment of fibrosis could be better with Fibroscan^®^ VCTE than with Fibrotest^®^ [59]. Of particular interest is the ELF test based on three proteins implicated in the remodeling of the extracellular matrix (i.e., amino-terminal propeptide of type III collagen (PIIINP), hyaluronan (HA), and tissue inhibitor of metalloproteinase 1 (TIMP-1)). Nobili’s study showed in 112 adolescents with MAFLD and an average age of 13 years that the ELF test assessed liver fibrosis stage with high sensitivity and specificity, better than those reported for adults [60]. In other words, the use of serum biomarkers directly reflecting the hepatic extracellular matrix changing appears to be a relevant and reliable noninvasive method for detecting fibrosis in pediatric MAFLD.

##### Liver Stiffness Measurement

Transient elastography, VCTE. This noninvasive method was validated for the assessment of moderate to severe liver fibrosis in pediatric MAFLD [52]. However, the size of the cohort was small (*n* = 54) and only Asian children were included. Different sizes of FibroScan^®^ probes according to the morphology of the patient are available. The S probe should be more appropriate to the thoracic perimeter of children with obesity. Current evidence supports the use of Fibroscan^®^ for the evaluation of MAFLD in the pediatric population [61].Magnetic resonance imaging, magnetic resonance elastography (MRE). MRE could more accurately detect advanced fibrosis because this technique is less influenced by abdominal adiposity and assesses a much larger liver volume. A recent study has showed an interest in the use of this imagery. However, this pediatric cohort was limited by the very low prevalence of advanced fibrosis [62].Point-shear wave elastography (pSWE). pSWE reveals precise sensitivity of marked fibrosis and has high success rates in discriminating severe from low-grade fibrosis in pediatric liver diseases. It is even more marked in the more severe forms. Shear wave elastographic values were more correlated with liver fibrosis when cirrhotic liver tissue was compared with normal tissue. However, the validity rates for differentiating low-grade liver fibrosis from nonpathological liver tissue were inconsistent. These results should be considered with great caution because of the very small number of patients with MAFLD included in that study (*n* = 11) [63].Bidimensional shear wave elastography (2D-SWE). Values of liver stiffness on 2D-SWE are sensitive to levels of fibrosis and necroinflammation. This technique can ensure excellent diagnostic performance for assessing liver fibrosis in pediatrics. However, potential confounding factors, such as necroinflammatory activity or the level of transaminases should be considered. A recent study including a limited number of MAFLD (5 out of 30 patients) and young patients has been published but requires further investigations [64].

## 4. Potential Recommendations

### 4.1. Medical Follow-Up

Progression of MAFLD could be common in children. In a retrospective long-term follow-up study of 66 children with a mean follow-up of 6.4 years, four out of five children who had a follow-up biopsy showed progression to a more advanced fibrosis stage [65]. The importance of a coordinated care chain and, a fortiori, of support in a local network has been evaluated. The care provided by regional French networks for the prevention and care of pediatric obesity is associated with a reduction in BMI z-score (from 3.6 ± 1.0 at baseline to 3.3 ± 1.1 at the end) that persists after the period of multidisciplinary care [66]. Otherwise, obese children followed by an expert physician in obesity are likely to be different from those children (even with the same BMI) who have not been referred to an obesity clinical team. Importantly, small increases or decreases in BMI have been significantly associated with the progression or resolution, respectively, of hepatic steatosis at follow-up. In addition, BMI in late adolescence has been reported as a strong and independent predictor of severe liver diseases later in life [67].

At clinical examination, the presence of acanthosis nigricans may reflect insulin resistance, so the presence of increased waist circumference and hepatomegaly should be recorded. Patients with NASH are also at higher risk of atherosclerosis due to peripheral IR, proatherogenic lipid profile, oxidative stress and systemic inflammation. Thus, blood pressure evaluation, control and monitoring should be an integral component of the clinical management of children with MAFLD.

### 4.2. Therapeutic Management and Recommendations

#### 4.2.1. Nondrug Strategies

For pediatric MAFLD patients, BMI z-score reduction or stabilization of body weight through exercise and dietary therapy must be the basic and primary therapeutic approach. For instance, consumption of high-carbohydrate meals and drinks should be limited, especially those containing high fructose corn syrup, as recommended in the NASPGHAN guidelines. 

Lifestyle modifications to promote body composition changes via physical activity, fighting against sedentary behavior (limiting screen time) and diet remain the mainstay of pediatric MAFLD management. The reduction in central abdominal fat relative to subcutaneous fat may dampen the progression of MAFLD. Moreover, these interventions work in a synergistic manner by enhancing both hepatic and extrahepatic insulin sensitivity and were shown to restore insulin metabolism pathways in 108 children with biopsy-proven MALFD at baseline. Moreover, long-term lifestyle changes, evaluated in 39/108 children (24 months), improve liver histology characterized by biomarkers (i.e., ALT, AST, GGT, lipid profile and HOMA-IR index) and by specific NASH liver hallmarks (i.e., steatosis, lobular inflammation, hepatocyte ballooning) [68]. A recent meta-analysis, based on 14 clinical trials, evaluating the impact of supervised exercise interventions on obesity and hepatic functions in a cohort of 1231 obese youths aged between 7 and 18 (including 821 in the exercise group vs. 410 in the control group) demonstrated a significant reduction in visceral, subcutaneous and intrahepatic fat, as well as GGT activity (but no change in AST and ALT). Based on this study, exercise intervention (especially aerobic exercises) is effective in obese youth with MAFLD and should be recommended [69]. Although exercise is highly recommended for MAFLD treatment by NASPGHAN and the American Association for the Study of Liver Diseases (AASLD) [47], randomized clinical trials evaluating the impact of exercise versus usual care on pediatric MAFLD are still missing. Furthermore, the effects of different modalities of training (i.e., volume, duration, intensity and type) on pediatric MAFLD must be established to optimize this care. All MAFLD patients should perform aerobic exercise for at least 60 min per day and, preferably three times a week, do resistance training and moderate-vigorous intensity aerobic training (e.g., walking, cycling), which are recommended to ameliorate metabolic parameters and mobility. 

Prevention around young patients with MAFLD is important. Thus, family health is crucial to consider, to prevent and best manage chronic diseases. For example, the involvement of the whole family in eating better, moving more and having less sedentary activities is very important. As children often have their parents as role models, it is important to convince parents to adopt a healthy lifestyle for the whole family. Thus, each member of the family unit should converge towards a healthier lifestyle in a sustainable way. 

Secondly, every caregiver and parent of an overweight or obese child with a MAFLD is also a citizen of a society with multiple laws, rules, and cultures. Societal health is a key element to consider in the management of these chronic diseases. Various governmental and legislative actions have proven to be effective deterrents to limit the consumption of substances that are potentially harmful to health such as ultra-processed products. Some cities and countries have introduced a soda tax which has helped to reduce the consumption of these drinks [70]. A courageous fight against industrial lobbies is needed. Measures to protect consumers, who are influenced by opportunistic digital encounters on the web, generated by clever algorithms targeting consumption “habits”, should be strengthened. Fair information to consumers on the origin and composition of products purchased is important. An education of children at school including knowledge of healthy eating and physical activity is needed. 

#### 4.2.2. Pharmacological Approaches

Currently, there is no approved pharmacological treatment of MAFLD in adults. The therapeutic targets are focused on the correction of insulin resistance (e.g., pioglitazone and glucagon-like peptide 1 (GLP-1), analogues), the reduction of systemic inflammation (i.e., the sodium-glucose transport protein 2 (SGLT2) inhibitors) and the liver–gut crosstalk (e.g., FXR agonists, bacterial derived products, etc.). As a brief summary of clinical trials against NASH in adults (novel medications at various stages of investigation [19,71,72]) the main drug targets are: ketohexokinase, the rate-limiting enzyme of fructose metabolism (PF-06835919, phase IIa); the acetyl-CoA carboxylase ACC inhibitor, enzyme regulating lipogenesis and fatty acid oxidation (GS-0976, phase IIa); the specific or pan agonist of PPAR agonist targeting fatty acid oxidation, adipocyte differentiation and/or immune cell polarization; regarding PPARγ agonists, the second generation PPARγ agonist MSDC-0602K has fewer systemic side effects and pioglitazone has clinical limitations related to the risk of bladder cancer and/or aggravation of heart failure (phase IIb); the thyroid hormone receptor beta agonists regulating, at least, the mitochondrial fatty acid oxidation (MGL-3196 and VK2809, phase IIa); the fibroblast growth factor 19 analogue enhancing FGF receptor 4 promoted glucose utilization and fatty acid oxidation (NGM282 and pegbelfermin, phase IIa); the FXR agonist acting on the liver and gut (for example, the semisynthetic bile acid obeticholic acid mediated FXR activation, phase III); the analogue of incretin GLP-1 enhancing insulin sensitivity (such as liraglutide, phase III); SGLT-2 inhibitor decreasing kidney glucose reabsorption, (such as empagliflozin). 

To date, few studies have reported pharmacological approaches in pediatric MAFLD (Table 3). Among them, clinical trials in children treated with 500 mg of metformin twice daily for 48 to 96 weeks found it to be associated with an improvement of hepatocyte ballooning (in 38% of the cases) without significant reductions in ALT levels [71]. In addition, a randomized double-blind TONIC trial has evaluated the impact of 96 weeks of treatment with metformin or vitamin E in 173 youths between 8 and 17 years old. Metformin or vitamin E versus placebo did not improve serum ALT levels or liver histology [72]. Currently, there is no conclusive evidence that metformin should be used as a primary treatment option for pediatric NASH. Current clinical trials focus on losartan, an orally administered angiotensin II receptor antagonist, which is currently on the market to treat high blood pressure. This molecule has been proposed as a novel treatment of MAFLD by improving steatosis and fibrosis. Preliminary evidence (phase II) suggests that losartan treatment is safe over 8 weeks in children with MAFLD and supports consideration of larger studies to test its efficacy [73]. A clinical trial evaluating the antagonist of SGLT-2 has been initiated in children, but data are still lacking due to the difficulties of recruiting patients during the COVID-19 pandemic (NCT03867487). Elafibranor acts as a co-agonist of PPAR-α and PPAR-δ [74]. This molecule was to be assessed in children and adolescent population (8 to 17 years of age) with NASH for a three-month period (NCT03883607). However, the development of Elafibranor has recently been stopped after disappointing results obtained in adults [75]. GLP1 agonists are currently evaluated in pediatric MAFLD. Liraglutide, a GLP-1 agonist, which already obtained the marketing authorization for type 2 diabetes, displayed very interesting results in adults with MAFLD (phase 2 trial) [76]. A clinical trial assessing the effect of liraglutide recently started in obese youth with prediabetes or early type 2 diabetes (T2D) and MAFLD (NCT05067621). The results will have to be watched closely. A phase III clinical trial has recently started to evaluate the effects of semaglutide in obese adolescents (NCT03919929). Semaglutide has already provided interesting results in adults (NCT02970942). A common positive effect of GLP-1 agonists is the significant weight reduction achieved, which contributes at least in part to the improvement of liver damage in MAFLD. Moreover, GLP-1 agonist and SGLT-2i have demonstrated a protective effect on cardio-vascular events and renal complications in adults [77,78,79]. However, all these treatments present potential side effects in adults, particularly in some of them who have urinary tract infections such as SGLT2, so these complications should be particularly sought out in children.

From evidence-based practice guidelines, including the pharmacological manage-ment of MAFLD published by the American Association for the Study of Liver Disease (AASLD) in 2012 and a joint work in 2016 from three European associations with exper-tise in obesity, diabetes and liver diseases (EASO, EASD, and EASL, respectively) [47,80], prescribing a drug is not yet recommended in children or adolescents with MAFLD. In addition, a liver biopsy before any medication is recommended in youths to reduce the risk of iatrogenic hepatotoxicity. In conclusion, it is much better to recommend intensive and appropriate lifestyle modification as part of a coordinated care pathway.

#### 4.2.3. Surgical Measures

The use of bariatric surgery remains the last resort and is strictly regulated. “Uncomplicated” MAFLD is not an indication for bariatric surgery. In 2009, the International Pediatric Endosurgery Group published guidelines for pediatric bariatric surgery focusing on the clinical impact of surgery on the most common early-onset obesity-related diseases. In France, for example, surgery in minors requires multidisciplinary follow-up in a specialized obesity center with pediatric competence (i.e., physiological, psychological and organizational criteria are evaluated). This care is part of a process lasting at least one year. Good compliance is also an essential part of the care (French health authority and American Society for Metabolic and Bariatric Surgery) [81].

The efficacy of laparoscopic sleeve gastrectomy in adolescents with severe obesity and biopsy-proven MAFLD (*n* = 20) has been reported. At baseline, all adolescents had stage 2 fibrosis and 30% had NASH. At the second liver biopsy performed 1 year after surgery, NASH resolution and fibrosis reversal occurred in 100% and 90% of patients, respectively [82]. In a recent study, 64 obese adults underwent bariatric surgery and were followed for 5 years. Five years after surgery, NASH was resolved in 84% of patients and fibrosis decreased significantly compared with baseline [83]. These beneficial effects were perceived from the first year of follow-up. However, these results must be qualified because of the heterogeneity of the results of other prospective studies in adults [84]. Concerning young candidates for bariatric surgery, we lack data, and it is essential to remember that this therapeutic measure is not the first resort. Moreover, Chavez-Tapia et al. [85] concluded that randomized clinical trials are needed to definitively assess the benefits and harms of bariatric surgery as a therapeutic approach for youth patients with NASH.

## 5. Conclusions

The growing prevalence of childhood obesity is strongly associated with the development of comorbidities previously considered to be diseases only observable in adulthood, such as MAFLD. The health consequences associated with chronic childhood liver disease are likely to be severe in the long term. For example, the increasing number of liver transplant registrants among younger adults (35–55 years) is likely explained by the increase in youth-onset MAFLD [86].

MAFLD is characterized by insulin resistance and hepatic steatosis, which is heavily influenced by a sedentary lifestyle, hypercaloric diets, genetic susceptibility, and epigenetics. Nevertheless, the interplays between these mechanisms remain poorly understood in childhood. In fact, the natural history of pediatric MAFLD is not yet well established, while children with significant fibrosis are considered particularly at high risk of long-term liver damage due to their long lifespan. In adults, MAFLD is an independent risk factor for type 2 diabetes, cardiovascular diseases and mortality.

New noninvasive diagnostic tools to assess hepatic steatosis and fibrosis are being developed and are becoming more widely used in clinical practice (such as FIB-4). However, they should be used with caution due to the lack of validation in the pediatric population. More specific scores to predict fibrosis in children have been proposed including PNFI and PNFS. Moreover, the combined use of noninvasive PNFI and ELF tests could be used as a first-line approach to assess liver fibrosis. Algorithms/scores including several biomarkers and transient elastography could provide additional diagnostic accuracy and might avoid the need for diagnostic liver biopsies. However, diagnosing MAFLD is challenging. The screening of MAFLD in youth remains inconsistent and the development of more accurate and precise noninvasive screening tests remains an active area of investigation. To date, liver biopsy remains the reference to estimate MAFLD. Furthermore, screening for MAFLDs in children is even more complicated because the manifestations of these diseases are different from those in adults. Interestingly, for example, the difference in the pediatric population is a pericentral steatosis that is more closely related to steatohepatitis and a periportal steatosis more related to liver fibrosis. Nevertheless, the mechanisms underlying the differences between the adult and pediatric steatohepatitis patterns have yet to be fully elucidated. 

Unhealthy eating behaviors and a sedentary lifestyle are recognized risk factors for MAFLD; thus, a balanced diet and regular physical activity play a key role in its treatment (Figure 5). A time of exchange with a young person must be anchored during care to define their needs and obviously take into consideration their desires so that they take pleasure in their physical practice and maintain it in the long term. These preventive measures are the cornerstone of the management of childhood obesity. This requires education of the various family members, of which the general practitioner traditionally has a global vision. The general practitioner is a key player in coordinating and relaying information and deciding on the medical and health actions to be initiated (e.g., preparation for pregnancy, regular monitoring, information on infant nutrition, promotion of sports activity in children, monitoring of growth curves, etc.). Many initiatives exist to combat childhood obesity, for example in France with the network for the prevention and management of pediatric obesity (RePPOP network, http://www.cnreppop.com, last visit: 12 February 2021) and in Europe through actions highlighted within the European Childhood Obesity Group (ECOG).

In addition, environmental health has become a priority for many cities throughout the world and for various elected representatives, for example, by promoting healthy food in public catering particularly from a very young age, through urban planning to promote physical activity and nonpolluting and environmentally friendly public transport. These preventive actions should also be focused on countries undergoing economic transition, particularly in some African countries and in India. In these countries, rapid changes in dietary habits lead to the concomitant presence of undernutrition and overnutrition defined as a “double burden of malnutrition” [70,87]. 

## Figures and Tables

**Figure 1 biomedicines-09-01915-f001:**
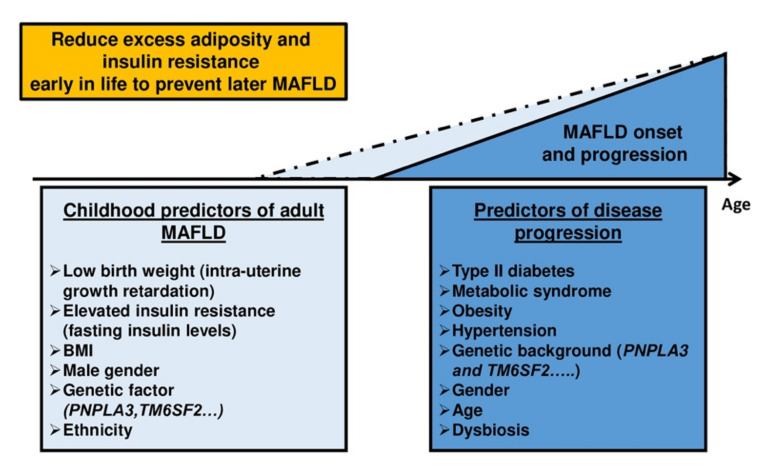
Potential MAFLD onset and progression among the youth population. Pediatric onset is likely more aggressive. MAFLD is a chronic liver disease that does not only affect adults. More and more children develop hepatic steatosis or even decompensated cirrhosis. These two populations nevertheless share some common predictors (e.g., BMI, genetic factors, insulin resistance, etc.) and disease progression (e.g., type 2 diabetes, obesity, hypertension, dysbiosis). Early treatment undeniably helps to prevent the risk of developing MAFLD (primary prevention) or of seeing it worsen (tertiary prevention). No pharmacological drug has yet been approved. Lifestyle measures seem to be the keystone treatment, so the diagnosis of the disease prior to the development of end-stage liver disease (i.e., severe cirrhosis step) is essential.

**Figure 2 biomedicines-09-01915-f002:**
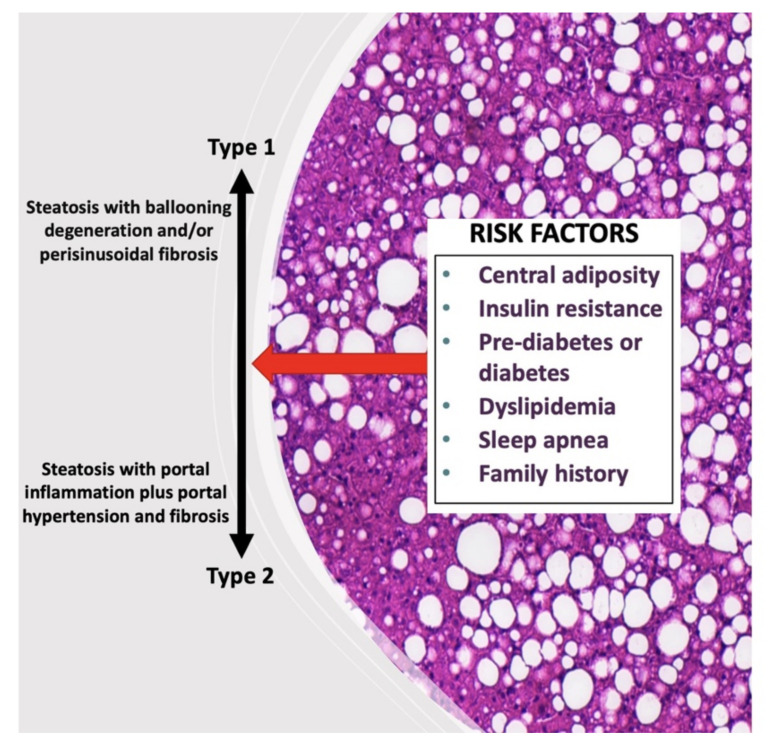
Typical histological appearance of pediatric MAFLD. Differences in the pediatric population have been reported compared to adults. The typical pattern in adults (named type 1 NASH) display the presence of steatosis with hepatocellular ballooning and/or perisinusoidal fibrosis (zone 3 lobular contribution) and relative sparing of the portal tracts. Pediatric NASH (called type 2 NASH) has been defined by the presence of steatosis with portal inflammation and/or fibrosis without perisinusoidal fibrosis and hepatocellular ballooning. However, the overlapping features of both types often occur in pediatric population.

**Figure 3 biomedicines-09-01915-f003:**
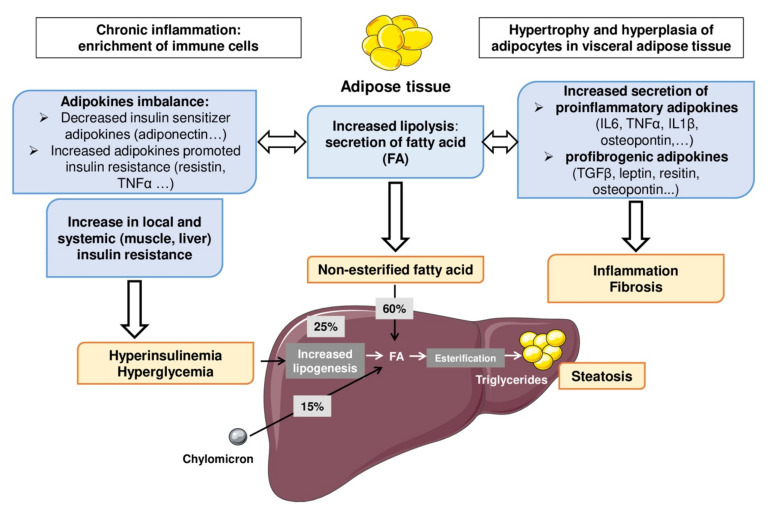
Adipose tissue–liver axis in the physiopathology of MAFLD. When the capacity of expansion of the subcutaneous adipose tissue is reached (i.e., hypertrophy and hyperplasia), an increased fatty acids (FAs) mobilization arises, resulting in visceral and ectopic fat deposition such as in the liver. Low grade chronic inflammation of adipose tissue plays a key role in this increased adipose tissue lipolysis and the onset of IR and type 2 diabetes. Hyperinsulinemia and hyperglycemia also participate to the development of hepatic steatosis by increasing liver lipogenesis. The steatosis–steatohepatitis transition could be dependent on lipid and glucose toxicity (release of DAMP) and increased proinflammatory and profibrogenic adipokines. These mechanisms involving the adipose tissue–liver axis are potentially involved in the appearance or even the aggravation of pediatric MALFD. Legend: FA, fatty acid; IL, interleukin; TGF, transforming growth factor; TNF, tumor necrosis factor.

**Figure 4 biomedicines-09-01915-f004:**
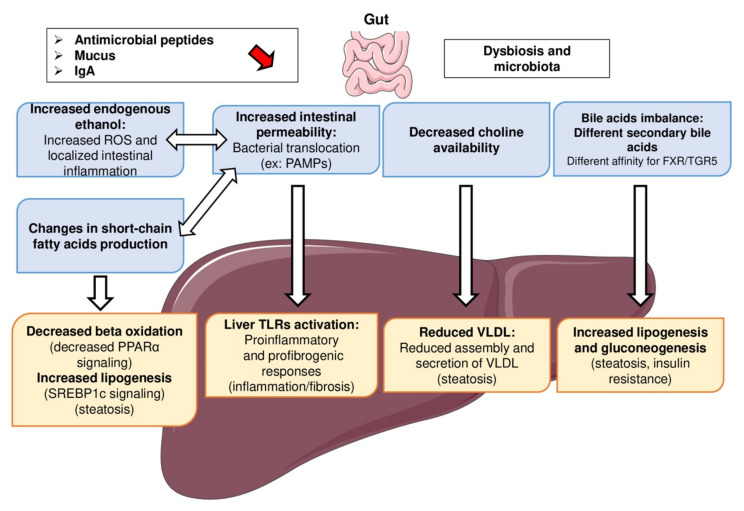
Gut–liver axis in the physiopathology of MAFLD. Many processes emanating from the gut’s function are involved in liver complications. An altered gut microbiota is the cornerstone of gut–liver axis disruption in the physiopathology of MAFLD. A high-fat diet has direct effects on gut microbiota composition. Abnormalities in intestinal microbiota mediate changes in secondary bile acids and subsequent TGR5/FXR signaling (increased lipogenesis and gluconeogenesis), gut inflammation and permeability (increased blood bacterial products (PAMPs) and hepatic proinflammatory and fibrotic responses), and gut microbiome-derived products such as choline availability, short-chain fatty acid and endogenous ethanol production (decreased beta oxidation and VLDL assembly, increased oxidative stress, respectively). These mechanisms involving the gut–liver axis are potentially involved in the appearance or even the aggravation of pediatric MALFD. Legend: FA, fatty acid; PAMPs, pathogen-associated molecular patterns; TLRs, toll-like receptors; PPAR, peroxisome proliferator-activated receptor; ROS, reactive oxygen species; SREBP, sterol regulatory element-binding protein; VLDL, very low-density lipoprotein.

**Figure 5 biomedicines-09-01915-f005:**
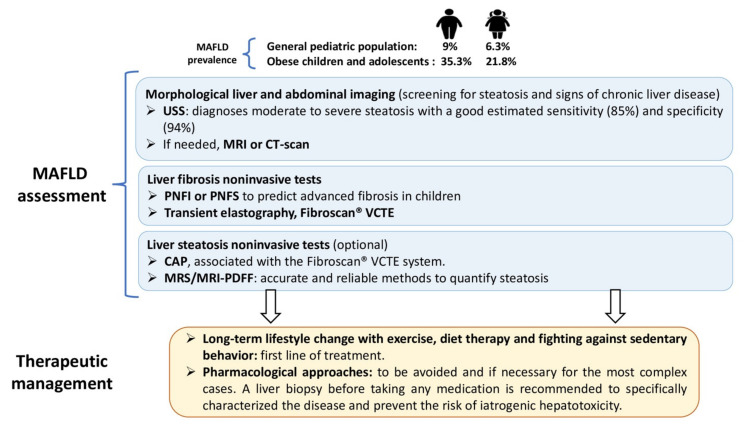
Treatment of children with MAFLD. Caring for a young person with MAFLD requires a multidisciplinary care path. People with MAFLD in childhood are at greatest risk for more severe complications later in life. The first step is to screen for the degree of severity of liver damage by noninvasive measures. MAFLD screening in youth remains inconsistent and the development of more accurate and precise noninvasive screening tests remains an active area of investigation. Liver analysis by imaging seems to be preferred. It is essential to differentiate imaging techniques evaluating steatosis (i.e., USS and CAP) from those assessing fibrosis (i.e., transient elastography VCTE, pSWE and 2D-SWE). Specific scores to predict fibrosis in children have been proposed, including the Pediatric NAFLD Fibrosis Index (PNFI) and Pediatric NAFLD Fibrosis Score (PNFS). An effective and preventive screening strategy to identify the children with MAFLD is needed. Combining these different diagnostic techniques seems to be the most appropriate. The second step is to perform a biopsy if there is a medical indication. Based on the NASPGHAN guidelines, further evaluation is necessary if the ALT levels are ≥80 U/L at the first visit, if ALT levels are persistently elevated (>3 months) to twice the upper limit of normal ALT levels (in girls ALT ≥ 44 U/L, in boys ALT ≥ 52 U/L) and/or the presence of type 2 diabetes. Interestingly, it has been reported that the BMI z-score (also called BMI standard deviation score, a measure of relative weight adjusted for child age, height, and sex) positively correlated with ALT serum concentrations. The primary issue is to make lifestyle modifications to improve diet, increase physical activity and fight against sedentary behavior (e.g., limiting screen time). These are recommended as the first-line treatment for all children with MAFLD. Avoidance of sugar-sweetened beverages is recommended as a strategy to decrease adiposity. Increasing moderate- to high-intensity physical activity and limiting screen time activities to <2 h per day is recommended for all children including those with MAFLD. Furthermore, the effects of different modalities of training (i.e., duration, type, intensity and volume) on pediatric MAFLD need to be established to optimize this care.

**Table 1 biomedicines-09-01915-t001:** Major single nucleotide polymorphisms identified in pediatric MAFLD. Single nucleotide polymorphisms (SNP) related to variants in *PNPLA3*, *TM6SF2*, *GCKR* and *IFNL4* and have a moderate impact on the risk of MAFLD because these depend on multifactorial interactions, and their impact on the pediatric MAFLD risk is currently not well defined.

Gene	Variant	Function	Clinical Form	Reference
*PNPLA3*	I148M	Lipid droplets remodeling	NAFL with the risk of progressive steatosis	[33]
*TM6SF2*	E167K	Impact VLDL-mediated lipid secretion leading to fat accumulation in liver	NAFL	[32]
*GCKR*	rs1260326	Inhibition of glucokinase enzymatic activity and modulation of hepatic lipogenesis	NAFL	[32]
*IFNL4*	rs368234815 δG	Increase fat accumulation	NAFL	[16]

**Table 2 biomedicines-09-01915-t002:** Noninvasive hepatic fibrosis test proposed in children. AST to platelet ratio index (APRI); fibrosis-4 (FIB-4) index; Pediatric NAFLD Fibrosis Index (PNFI); Pediatric NAFLD Fibrosis Score (PNFS). TG, triglyceride; WC, waist circumference.

Indexes Scores	Clinical and Biological Data								
	Age	BMI	WC (cm)	Fasted TG	ALT	AST	Platelet Count (10^9^/L)	Albumin (g/dL)	Reference
**APRI**					X	X	X		[57]
**FIB-4**	X				X	X	X		[57]
**PNFI**	X		X	X					[58]
**PNFS**	X	X			X	X	X	X	[53]

**Table 3 biomedicines-09-01915-t003:** Main pharmacological molecules whose research is in progress or suspended for the treatment of adult versus pediatric MAFLD. In adult MAFLD, ketohexokinase (KHK), the rate-limiting enzyme of fructose metabolism (PF-06835919); the acetyl-CoA carboxylase ACC inhibitor (GS-0976), an enzyme that regulates lipogenesis and fatty acid oxidation, however, clinical development of the ACC inhibitor was terminated owing to reduced platelet counts; the specific or pan agonists of PPAR, agonists that target fatty acid oxidation, adipocyte differentiation and/or immune cell polarization, though this did not exhibit a statistically significant effect on NASH resolution and the second generation PPARγ agonist (MSDC-0602K) has fewer systemic side effects than pioglitazone, which has clinical limitations reported to the risk of bladder cancer and/or the aggravation of heart failure; the thyroid hormone receptor beta agonists that regulate, at least, mitochondrial fatty acid oxidation; the analogue of incretin glucagon-like peptide 1 (GLP-1) that enhances insulin sensitivity (liraglutide); and the sodium-glucose transport protein 2 (SGLT-2) inhibitor that decreases kidney glucose reabsorption (empagliflozin). In pediatric MAFLD, losartan has been proposed by improving steatosis and fibrosis; SGLT-2 is in progress; semaglutide in obese adolescents is also in progress; Elafibranor acts as an agonist of PPAR-α and PPAR-β/δ, the results are not yet published; the use of liraglutide in obese youth with prediabetes or early type 2 diabetes (T2D) and MAFLD has just started.

Clinical Trial	State	Sample and Size	Drug	Target	Potential Side Effects
NCT01913470	Phase IIa	11–19 years old N = 12	Losartan	Angiotensin II receptor antagonist	No specific side effect reported
NCT03883607	Phase IIa	8–17 years old N = 10	Elafibranor	PPAR-α and PPAR-δ	No specific side effect reported
NCT03867487	Phase IIa (suspended due to COVID-19 pandemic)	12–20 years old N = 40 (estimated enrollment)	Empagliflozin	SGLT-2	In adult, urinary tract infections
NCT05067621	Phase III (ongoing patient recruitment)	10–21 years old N = 60 (estimated enrollment)	Liraglutide	GLP-1 receptor agonists	Gastrointestinal symptoms
NCT03919929	Phase III (ongoing patient recruitment)	12–21 years old N = 50 (estimated enrollment)	Semaglutide	GLP-1 receptor agonists	Gastrointestinal symptoms

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
