# Peer review of "Metabolic Fatty Liver Disease in Children: A Growing Public Health Problem"

_biomedicines, 2021, doi:10.3390/biomedicines9121915_

Round 1

Reviewer 1 Report

I read with great interest the paper “Metabolic fatty liver disease in children: a growing public health problem" by Le Garf et al.

 The article is well written, and only minor spell check is needed. The paper has a good design. The article is logically divided into sections and subsections. There are several tables and figures of good quality. The references cited are relevant and adequate. Some sections, though, should be expanded.

Major comments:

  1. “2.4. Intra- and extrahepatic mechanisms” section must be improved as it is too summarized (DOI: 3390/pr9010135; DOI: 10.31083/j.rcm2203082).
  2. “4.2.2 Pharmacological approaches” as all trials are ongoing and few data about efficacy are available, it would be advisable also to talk about safety. In fact, one of the main causes of discontinuation in adults of SGLT2i are urinary infections, which, I suggest there would be even more in children, according to their age.
  3. 608-609: not only semaglutide, but it is a drug class benefit. Moreover, it also interest SGLT2i (DOI:3390/ijms22115863).

Author Response

We thank the reviewer for his/her important comments

  • “2.4. Intra- and extrahepatic mechanisms” section must be improved as it is too summarized (DOI: 3390/pr9010135; DOI: 10.31083/j.rcm2203082).

We have now expanded this section and included additional reports, including recommended articles.

Caturano et al., 2021: cited lines 181-192

Galiero et al., 2021: cited lines 194-197

  • “4.2.2 Pharmacological approaches” as all trials are ongoing and few data about efficacy are available, it would be advisable also to talk about safety. In fact, one of the main causes of discontinuation in adults of SGLT2i are urinary infections, which, I suggest there would be even more in children, according to their age.

We have now clarified these important details in Table 2 and in the main manuscript (Lines 501-502): please see the attachment. 

  • 608-609: not only semaglutide, but it is a drug class benefit. Moreover, it also interests SGLT2i (DOI:3390/ijms22115863).

Moreover, GLP-1 agonist and SGLT-2i have demonstrated a protective effect on cardio-vascular events and renal complications in adults (Toyama et al., 2021; Pasternak et al., 2020; Brown et al., 2021).

  • Lines 669-673

However, all these treatments present potential side effects in adults, particularly in some of them who have urinary tract infections such as SGLT2, so these complications should be particularly sought out in children.

  • Table 3 (Line 692)

Reviewer 2 Report

This is a well written review that focuses on the development of MALD and NASH in the pediatric population. The review includes all of the important topics and contributes to the field.

Minor:

Line 334 define NASPGHAN

Line 368 define BMI Z score

Line 434 define how the index and score are determined

Author Response

We are delighted that the reviewer appreciated our work. Below are the responses to the minor comments Minor comments:

  • Line 334 define NASPGHAN

We have now clarified the NASPGHAN definition (Executive Council of the North American Society for Pediatric Gastroenterology, Hepatology, and Nutrition (NASPGHAN)), lines 380-381.

  • Line 368 define BMI Z score

We have now defined the BMI Z-score (also called BMI standard deviation score, a measure of relative weight adjusted for child age, height and sex), lines 413-414.

  • Line 434 define how the index and score are determined

We have now indicated the different clinical and biological data necessary to calculate the different scores or indexes of hepatic fibrosis (cf. line 502): please see the attachment.

Round 2

Reviewer 1 Report

I am satisfied with authors answer. In my opinion, the paper can be further process for publication.